# Brain Fog and Fatigue following COVID-19 Infection: An Exploratory Study of Patient Experiences of Long COVID

**DOI:** 10.3390/ijerph192315499

**Published:** 2022-11-23

**Authors:** Emily E. Chasco, Kimberly Dukes, DeShauna Jones, Alejandro P. Comellas, Richard M. Hoffman, Alpana Garg

**Affiliations:** 1Institute for Clinical & Translational Science, University of Iowa, Iowa City, IA 52242, USA; 2Center for Access & Delivery Research and Evaluation (CADRE), Iowa City Veterans Affairs Health Care System (ICVAHCS), Iowa City, IA 52246, USA; 3Department of Internal Medicine, Division of General Internal Medicine, University of Iowa, Iowa City, IA 52242, USA; 4Department of Community and Behavioral Health, College of Public Health, University of Iowa, Iowa City, IA 52242, USA; 5Department of Internal Medicine, Division of Pulmonary, Critical Care and Occupational Medicine, University of Iowa, Iowa City, IA 52242, USA; 6Holden Comprehensive Cancer Center, University of Iowa, Iowa City, IA 52242, USA

**Keywords:** post-acute sequalae of SARS-CoV-2 (PASC), qualitative, patient experience, brain fog, fatigue, quality of life, long COVID/post-COVID-19 syndrome, mental health, occupational, social, economic impact

## Abstract

Post-acute sequelae of SARS-CoV-2 (PASC) is a poorly understood condition with significant impact on quality of life. We aimed to better understand the lived experiences of patients with PASC, focusing on the impact of cognitive complaints (“brain fog”) and fatigue on (1) daily activities, (2) work/employment, and (3) interpersonal relationships. We conducted semi-structured qualitative interviews with 15 patients of a Midwestern academic hospital’s post-COVID-19 clinic. We audio-recorded, transcribed, and analyzed interviews thematically using a combined deductive-inductive approach and collected participants’ characteristics from chart review. Participants frequently used descriptive and metaphorical language to describe symptoms that were relapsing-remitting and unpredictable. Fatigue and brain fog affected all domains and identified subthemes included symptoms’ synergistic effects, difficulty with multitasking, lack of support, poor self-perception, and fear of loss of income and employment. Personal relationships were affected with change of responsibilities, difficulty parenting, social isolation, and guilt due to the burdens placed on family. Furthermore, underlying social stigma contributed to negative emotions, which significantly affected emotional and mental health. Our findings highlight PASC’s negative impact on patients’ daily lives. Providers can better support COVID-19 survivors during their recovery by identifying their needs in a sensitive and timely manner.

## 1. Introduction

The COVID-19 pandemic has drastically affected the lives of millions of people around the globe [1]. Besides significant mortality, there is a growing population of patients with post-acute sequelae of SARS-CoV-2 (PASC), a term coined by the National Institute of Health (NIH) and commonly referred to as long COVID or post-COVID-19 condition [2,3]. The Centers for Disease Control and Prevention (CDC) defines post-COVID-19 conditions as “new, returning, or ongoing health problems people can experience after first being infected with the virus that causes COVID-19” [4]. Studies on long COVID have reported multisystem involvement with a range of physical and neuropsychiatric symptoms after infection [5,6,7,8,9,10]. Shortness of breath, cough, chest pain, dizziness, fatigue, anosmia, dysgeusia, headache, and cognitive difficulties are among the most common symptoms reported in the studies [5,6,8]. Similar symptoms including fatigue, concentration difficulties, mood changes, and sleeping difficulties have also been reported in children and adolescents even after asymptomatic initial infection [11,12,13]. These symptoms can significantly impact quality of life for COVID-19 survivors [14,15,16]. Several initiatives have been launched to gather clinical data and track symptoms and outcomes of COVID-19 [17,18].

Early qualitative research on the broader patient experience of COVID-19 has revealed some common themes including symptom variability, a heavy sense of stigma, siloed care, the burden on patients to access care, and fear of a permanent reduction in physical and cognitive abilities [19,20,21,22,23,24,25,26,27]. Open-ended survey studies have also highlighted themes such as multiple symptoms with variable intensity, challenges seeking medical care, motivation to move forward with self-compassion, and also interest in participating in research to explore more about the condition [10,28,29]. A recent meta-analysis of 81 studies reported 32% of patients experienced fatigue and 22% reported cognitive impairment 12 or more weeks after COVID-19 diagnosis [30]. Cognitive complaints (i.e., “brain fog”) and fatigue have been reported in the literature as two of the most debilitating symptoms for patients with PASC [31,32]. However, our understanding of how patients experience and make sense of these symptoms in their daily lives is still in early stages. It is important for providers to understand how patients themselves describe PASC and its impact on their lives to help patients manage their condition. Validation of patients’ concerns and deeper insight into their condition is needed beyond illness management. In this exploratory qualitative study, we aimed to better understand the lived experiences of patients with PASC with a particular focus on how patients describe cognitive complaints (“brain fog”) and fatigue, and the impact of these symptoms on daily activities, work/employment, and interpersonal relationships.

## 2. Materials and Methods

### 2.1. Participants and Setting

Study participants included adult patients of the University of Iowa (UI) Hospitals and Clinics Post-COVID-19 Clinic who were able to speak and understand English, and who had persistent health concerns more than 3 months after SARS-CoV-2 infection. The UI Institutional Review Board approved all study activities (IRB #202103489) and patients consented prior to participation.

### 2.2. Recruitment and Data Collection

The principal investigator (AG) reviewed the post-COVID-19 clinic’s registry and purposively selected patients eligible for participation based on the presence of persistent (>3 months after infection) health concerns. Purposive sampling is a frequently used strategy in qualitative research, in which individuals with in-depth knowledge of or relevant experience with the phenomenon of interest are identified for recruitment. Its strengths lie in the “selection of information-rich cases for the most effective use of limited resources,” and it was an appropriate choice for a small exploratory qualitative study such as this [33,34]. Thirty patients were recruited by email to participate in semi-structured phone interviews (one per participant); fifteen agreed to participate. 

AG, an internal medicine physician with experience treating patients with PASC, and a qualitatively trained medical anthropologist (KD) developed an interview guide informed by AG’s clinical experience and the PASC literature. The guide included 13 questions as well as follow-up probing questions. We first asked participants to describe their illness since its inception, with a follow-up prompt asking what symptoms they had had following COVID-19 infection if needed. Subsequent questions asked about the experiences of everyday life (e.g., home and work/employment) with long COVID, interpersonal relationships and social support, experiences with healthcare and healthcare providers, treatment and recovery, and feedback for healthcare providers and policymakers. Questions were broadly worded and open-ended, to reduce social desirability or other bias, and interviewers also used additional probes to encourage participants to expand on their responses [35,36]. AG, KD, and a second qualitative researcher (DJ) interviewed patients from July–October 2021. 

### 2.3. Data Management and Analysis

We audio-recorded all interviews, which averaged 38.5 min (range 26–83 min) in length. Audio files were de-identified and transcribed manually by a professional transcription service. We imported transcripts into MAXQDA version 20.4.1 [37] for analysis from November 2021–January 2022. The interdisciplinary team, including qualitative experts and a physician, conducted a rigorous, systematic thematic analysis [38]. EC, a qualitatively trained medical anthropologist on the study team, developed a preliminary codebook incorporating a priori deductive codes generated from the interview guide, the literature, and known PASC sequelae, and inductive codes based on an initial reading of 3 (20.0%) interview transcripts. The analysis team (AG, KD, EC) then independently coded 2 transcripts, and met after each to compare coding and refine the codebook. Discrepancies were resolved through team discussion. Finding strong consensus, the team repeated this process with an additional 2 transcripts (a total of 4 team-coded transcripts, or 26.7%) to increase reliability. EC applied the final codebook to the remaining 11 (73.3%) transcripts. The team met regularly to review questions and documented analytical decisions and emerging themes in memos. In this manuscript, we focus on findings related to the impact of fatigue and brain fog on daily activities, work/employment, and interpersonal relationships. 

## 3. Results

### 3.1. Demographics

Participants’ characteristics are reported in Table 1. The majority were females (66.7%) and received outpatient care (60%) during their acute illness. Initial onset of acute COVID-19 in participants occurred from February 2020–February 2021, with an average length of symptoms (i.e., time from diagnosis to study participation) of 11 months (range 6–19 months). Participants who acquired the virus early in the pandemic reported they could not always access testing promptly. Ultimately, 14/15 (93.3%) participants received a confirmed COVID-19 diagnosis (which was strongly suspected based on clinical features of the unconfirmed participant). During the acute phase, 6 (40.0%) participants were hospitalized, 2 (13.3%) of which were admitted to the intensive care unit.

### 3.2. Descriptive and Metaphorical Language for Symptoms

Participants attributed a variety of symptoms to PASC, and highlighted brain fog and fatigue as particularly salient. However, they often described their experiences using descriptive and metaphorical language rather than specific symptom names. Appendix A highlights the language and descriptions used for fatigue, brain fog, and other symptoms. For example, participants frequently referred to “weakness” or instability during physical exertion in discussions of fatigue. Another explained brain fog in technological terms, saying, “And it’s messing receptors up, like a wi-fi connection to a router…that connection’s not there...” PASC affected participants physically, mentally, and emotionally in terms of how they viewed themselves and their ability to be fully present in their lives. Their words frequently emphasized a sense of loss, either of their previous identity or of an imagined future self. A middle-aged participant said, “It’s like my body has aged twenty years.” In the remainder of this section, we focus on findings related to the impact of brain fog and fatigue on three domains: daily activities, work/employment, and interpersonal relationships (with supporting illustrative quotations reported in Appendix A). 

### 3.3. Impact on Daily Activities

#### 3.3.1. Running a Household

Brain fog and fatigue affected participants’ ability to complete household chores (e.g., cleaning, cooking, grocery shopping), as well as personal tasks (e.g., eating, exercising) and leisure activities (e.g., reading, gardening). Many stressed that fatigue and perceived related symptoms (e.g., weakness, dizziness, shortness of breath) made the mundane tasks of running a home more challenging. Participants described difficulty standing for long periods or using stairs, with some needing to hold on to something for stability. References to frequent naps or rest periods were also common: “There have been days when I get up in the morning and eat breakfast and go back to bed, then get up and eat lunch and go back to bed”. Participants also cited intermittent headaches, pain, and sleep issues (e.g., insomnia, oversleeping) as impacting their daily activities. If they were able to complete household chores, the costs of these activities were exhaustion, sleepiness, and the need for additional rest. Participants stated they were unable to complete as many tasks as they typically had pre-COVID-19. 

#### 3.3.2. Relapsing-Remitting and Unpredictable Symptoms

All participants described symptoms (and their severity) as changing throughout the day. Participants frequently stated they had more energy in the mornings, using that time to accomplish basic household tasks (e.g., laundry, dishes) to “make some contributions to the household”. Brain fog and fatigue typically worsened as the day went on. For example, participants described needing to sit down or experiencing more “muddled thoughts” in the afternoon. 

Relapsing-remitting symptoms over longer periods were also common, making it more difficult for participants to predict and manage their illness: “I think that you just can’t plan on it because you never know when—I have good days, and I have bad days”. Some recounted consecutive routine days followed by episodes during which they forgot what they were doing mid-task. One participant said that “some days I can’t tell you whether I gave my dog his medicine first thing in the morning, which I’m awfully careful about”; another shared the same experience about her own medication. The unpredictability of relapsing-remitting cycles was a major source of frustration and other negative emotions (e.g., anxiety, depression, fear, sadness) in participants: “Every time I think [my sense of taste is] starting to come back and—it goes again”. 

#### 3.3.3. Synergistic Relationship of Fatigue and Brain Fog

Importantly, many participants believed fatigue and brain fog to have a synergistic relationship (e.g., worsening brain fog with increasing fatigue), and features of brain fog (e.g., difficulties with memory, recall, or following a sequence) negatively affected participants’ ability to complete daily tasks even when they had the energy to do so. To cope with this synergy, some participants attempted to ration activity between periods of rest to maximize what they could accomplish. For example, one described centralizing her activities between the kitchen, bathroom, and living room: “I can handle it if I stay between…I don’t have to go too far, and I can sit down in any spot”. Rationing was challenging, however, particularly for working participants. Some had difficulty recognizing when they crossed a certain fatigue threshold, including a participant who shared that her husband encouraged her to rest following chores when she started “talking, he says, like gibberish, not making sense or something”. 

#### 3.3.4. Difficulty with Driving

Of note, nine (60.0%) participants referenced the effects of PASC symptoms on driving specifically. Participants typically attributed an inability to drive or discomfort driving to brain fog (e.g., memory or focus issues), fatigue, and/or increased anxiety. One drove into their garage and could not remember what to do next: “Do I turn [the car] off? Do I close the garage?”, while two forgot familiar destinations mid-drive. Some participants limited their driving to specific circumstances. For example, one drove in familiar rural areas, but not in more urban settings—where her medical appointments were located—because the increased traffic made it too difficult to focus. Reduced driving capability further restricted participants’ activities and increased their reliance on others. One participant mentioned, “I really have to prepare myself and lay my stuff out and, to know to get there to do, like I said, I didn’t drive for a long time. Now I’m driving myself. But you get anxious, anxiety. Everything’s at a slow pace”.

### 3.4. Impact on Work/Employment

#### 3.4.1. Employer Support

In this study, six (40.0%) participants reported current unemployment as a direct result of PASC symptoms; one (7.0%) participant retired pre-pandemic but stated she too would have been unable to work with her symptoms. Most participants felt supported by their employers during their acute infection, and even as they initially developed PASC. Support manifested as checking in with the participant (i.e., emotional support), approving sick/vacation time or leaves of absence, and accommodating medical appointments. Three participants were able to adapt work tasks to accommodate their symptoms (e.g., schedule changes, cheat sheets, breaks) due to self-employment or remote work. 

#### 3.4.2. Self-Perception of Ability to Work

Brain fog appeared to have the greatest impact on the work/employment domain for study participants, in part due to new struggles with previously easily managed tasks. Fatigue was a barrier in that participants often struggled to make it through the workday, as one noted, “I’ve missed a lot of work…I don’t get paid when I miss work, so it makes it very hard”. However, participants emphasized that brain fog not only impacted their employers’ perceptions of them, but also their perceptions of themselves as capable, experienced professionals, able to trust their own skills and abilities. Participants reported forgetting how to do routine tasks or basic computer functions that they had done for years, mishandling communication, or having difficulty speaking. They used phrases such as “spaced out”, “losing it”, and “dropping balls” to describe their work functioning. 

#### 3.4.3. Difficulty with Multitasking

Most participants had difficulty with multitasking and focus, perceiving themselves to be less efficient and productive: “I’m one of those people who multitask. I don’t forget things… [but now], I cannot multitask as well. I still can, but it’s not—like I’m doing a bunch of things okay. I’m not doing a bunch of things great”. Examples included missing email communication, struggling to learn and incorporate new tasks, and forgetting to complete required documentation. In response, some took more time to complete tasks, or completed fewer tasks each day; others implied their work environments were fast paced in ways that made it difficult or impossible to accommodate their new limitations. 

#### 3.4.4. Colleagues’ Perceptions of Participants’ Ability to Work

Furthermore, the severity of symptoms made it difficult for participants to hide their condition in the workplace. As a result, participants struggled with their own expectations and those of others: “My IQ scores are high and so people expect a lot of me at work…and all of a sudden, I just can’t remember things”. Participants recounted asking colleagues for help on “simple stuff” and finding problem-solving and reading instructions challenging. Some reported involuntary physical responses to symptoms or using coping mechanisms such as closing their eyes, sitting down, or putting their head down, which were visible to colleagues. Managing their PASC symptoms together with perceptions of their illness’ severity or legitimacy took a significant mental toll on participants over time.

#### 3.4.5. Stigma at Work

Participants believed their colleagues did not understand the relapsing-remitting nature of symptoms. One described a colleague’s comment during a period when she averaged one morning off per week due to extreme fatigue: “Well, I saw you last Saturday and you didn’t look too bad then”. Related to this was the concern that employers doubted the legitimacy of PASC or the recovery process. Many responses, even from participants who felt supported in the workplace (e.g., through accommodation), indicated they perceived their performances were scrutinized and that this scrutiny had implications for their continuing employment. Some took a more pragmatic approach to symptom management in order to remain in the workplace: “I’m just gonna do what I can. If I need to lay my head down, I need to lay my head down”. However, others found it difficult or impossible to continue in their position (or even career) when it became clear there would be no quick resolution to their brain fog and fatigue. 

#### 3.4.6. Concerns about Unemployment/Loss of Income/Benefits

Unemployed participants recognized that loss of income and benefits burdened their partners and families. One participant illustrated this: “…I don’t have a job and my [partner], unfortunately…works two jobs now”. Most were able to rely on a partner’s income and support in the short-term, but were concerned for the long-term. Highlighting how this impacted their healthcare, a participant said, “…it takes money for the travel and then to pay for what insurance doesn’t cover is expensive. Financially, it’s draining, which of course, again, makes it stressful”. Both employed and unemployed participants stressed that COVID-19 survivors needed more long-term support, including recognition of PASC as a legitimate medical condition, eligibility for disability benefits, and policy action at the state and federal levels. In their absence, it fell to participants—who were also managing significant PASC symptoms—to navigate often complex insurance and disability benefit systems.

### 3.5. Impact on Interpersonal Relationships

#### 3.5.1. Loss/Change of Roles

Participants reported that not only brain fog and fatigue, but also pain (e.g., headache, joint, muscle) and mental health symptoms (e.g., depression) greatly impacted their ability to fulfill their accustomed relationship role(s). Within the household, these roles included family member, partner, caregiver, and/or provider, and the same symptoms that made it difficult to complete daily tasks often left participants feeling inadequate or isolated from the family core. Describing this phenomenon, a participant said a good day was one where “I can recognize I’m getting tired and it’s time for me to be done, as opposed to getting sent upstairs’ cause I fell asleep…that I can be part of the family with minimal pain…”. 

#### 3.5.2. Effect on Partners

Overall, participants in romantic relationships felt supported by their partners, but were concerned about the future, given their slow recovery. One confessed, “I hope that my family doesn’t grow tired of me being tired all the time…”. Again, participants voiced an acute awareness of the toll that their condition took on partners, who were often cast in a caregiver role. Their shared lives, routines, and hobbies also frequently looked quite different with PASC: “We always have done stuff ever since we been together…like we go to an event, or we go visiting friends or family, we always go together…we could do four, five things in a day. And now…it takes all day just to do one thing”. Some partners developed burnout or other stress-related conditions, as a participant shared, “My [partner] has developed anxiety and had to go on medication… [my partner has] taken on a lot of responsibility”. 

#### 3.5.3. Effect on Parenting

Participants with children at home frequently stated it was difficult to be present as a parent. When describing lunch with his daughter on a bad day, a participant said, “I try to interact with her, but I don’t have the energy to do anything and then I end up just going back upstairs and it feels like a waste”. Nearly all participants attempted to pace or ration energy to complete daily tasks, including parenting. One had been pregnant during the pandemic; currently unemployed, she struggled to manage household tasks and care for her child. A working parent shared, “When [my kids] wanna do activities and stuff, it’s hard for me to find the energy, cause I feel like I spend more of my energy just trying to get through the day at work”. Others spoke of relying on older children at home to shoulder more responsibilities and feeling guilty as a result. Some participants received emotional or tangible (e.g., rides) support from their adult children, but also cited difficulty holding conversations with or traveling to their children.

#### 3.5.4. Effect on Relationships Outside of Household

Within the context of relationships with extended family, friends, neighbors, and colleagues, the impact was more varied. Some participants felt they and their families had received significant emotional and practical support during their illness, in the form of meals, gift cards, rides, and household assistance. Two described siblings stepping up to care for aging parents when they were unable to continue. However, stories of waning communication or fractured relationships were also common. Some participants shared that others believed them to be inventing or exaggerating their symptoms, particularly if those individuals had experienced few COVID-19 cases in their social networks or those cases had resolved following the acute stage. Proximity to PASC was perceived to mitigate these beliefs. As one participant said, her sibling “…doesn’t see me on a daily basis, so [(s)he] doesn’t see [the] full effect”. Some also felt their symptoms and altered state made others uncomfortable or unsure of how to continue the relationship. A participant poignantly stated, “It takes a lot more energy for [friends] to look at someone their own age, and it looks like they’re failing. They don’t wanna look at someone like that”. 

#### 3.5.5. Feelings of Isolation

Feelings of isolation were exacerbated in relationships outside the household due to both internal and external causes. Internal causes included mental and physical symptoms that participants reported made it more difficult to leave home, take part in “normal” activities, and converse with others. One described having to remain inside while extended family enjoyed outdoor recreation during a camping trip. Another no longer had the stamina to volunteer. Despite perceptions shared above, participants more commonly attributed isolation to their own behavior changes rather than to others’, or to their own perceptions that others would not accommodate their interactions. Participants frequently attached feelings of guilt and shame (e.g., being “a bad friend”) to these stories, for example, a participant withdrew when symptoms were at their worst and only later learned her friend died in the interim. As in the household and the workplace, some participants found it difficult to cope with upended interpersonal dynamics. A participant who described himself as “a help provider, not a help receiver, by nature”, shared, “I’m limited in who I’m willing to let come over and see me in this state”.

#### 3.5.6. Stigma and Difficulty Being Believed

Participants were cognizant of broader narratives (e.g., conspiracy theories, misinformation) around both COVID-19 and chronic invisible illnesses generally and experienced both perceived and explicit stigma, as noted above. Participants made clear that these narratives, present within individual relationships and wider communities, formed a harmful “subtext” and contributed to their isolation. Some felt under scrutiny when with others or pressured to “fit into the things that people are saying” about valid PASC versus non-specific symptoms. Relapsing-remitting and unpredictable symptoms were also frustrating because others responded with skepticism and confusion when “everything’s okay one day and then the next day you’re not”. To participants, some relationship breaches felt permanent. For example, one said their sibling was “dead to me”, because the sibling “…still believes it’s fake…that I almost died because I’m scared of everything, and that I only had the flu…”. 

#### 3.5.7. Changing the Perceptions of Others

While less common, a few participants believed their own experience had caused others to reconsider their skepticism. For example, one participant felt their experience changed the perspective of community members “that still believe…that it’s a hoax…I’ve grown up here and they’re like, ‘Oh my gosh, [participant’s] goin’ through this?’” This participant went on to say a friend convinced others to “get their COVID shots” by sharing the participant’s story. Regardless of how PASC had impacted their interpersonal relationships, participants stressed the need for the condition to be taken seriously.

## 4. Discussion

We conducted a small exploratory qualitative study to better understand the experiences and perceptions of patients with PASC. Participants’ use of descriptive and metaphorical language to portray brain fog and fatigue indicate that it is important for providers to understand *how* patients experience symptoms in their daily lives. Furthermore, we initially focused on PASC’s impact on three specific domains, yet in participants’ recounting these domains were not distinct, but rather fluid and overlapping. There was significant impact on daily activities (e.g., cooking, spending time with family) as well as work/employment (e.g., working efficiently or working full-time) and interpersonal relationships (e.g., close family members, colleagues at work, friends), with impacts in one domain often having implications for the others. 

Our findings align with other survey-based and qualitative studies that have described the experiences of patients with PASC. One UK-based qualitative study by Callan et al. included 50 participants and described the neurocognitive symptoms and impact on various domains including personal, professional identity, strategies for self-management, and challenges in accessing and navigating the healthcare system [20]. Heiberg et al. found survivors reported gratitude and new perspectives on their lives despite PASC-related symptoms; however, they focused specifically on older adults between 60–90 years of age [26]. Rushforth, et al. have compared long COVID illness narratives to other illness narratives, noting important similarities to chronic conditions and unexpectedly protracted illnesses [21]. In particular, they point to perceptions of not being believed and of what they term “collective action” on the part of patients to address their concerns, such as forming online communities, advocating for themselves and others, and sharing their stories.

In that vein, the findings of persistent debilitating neurocognitive impairment and fatigue also resonate with other conditions such as fibromyalgia (“fibro fog”), cancer-related cognitive impairment (“chemo brain”), traumatic brain injury, chronic fatigue syndrome, mast cell disorders, autoimmune disorders, and postural tachycardia syndrome [39,40,41,42,43,44,45,46,47]. In a recent systematic review, 20 selected studies were reviewed to highlight similarities and overlap between the symptomatology of PASC and myalgia encephalomyelitis (ME/CFS) [48]. Underlying precipitating factors for ME/CFS included infections, stressful incidents or major life events, and exposure to environmental toxins [49,50]. It is also interesting to note similarities between cognitive difficulties described in fibromyalgia and PASC, where participants described impairment in memory and difficulty when tasks are complex, and attention is divided [40]. Perceived cognitive dysfunction was further exacerbated by factors such as depression, poor sleep, anxiety, and pain [40].

Importantly, many patients in our study believed brain fog and fatigue to have a synergistic relationship (i.e., worsening brain fog with increasing fatigue). This has been described in the literature as well [51]. Fatigue has been well established in neuroinflammatory conditions such as multiple sclerosis [52]. With underlying autoimmune dysfunction and chronic persistent ongoing inflammation, further studies are needed to investigate the pathophysiology of fatigue and neurocognitive impairment in PASC [53]. This study was aimed at gathering patient experiences, but the symptomatology described by patients can also help guide development of research studies or protocols to better understand PASC’s underlying mechanisms. 

This study highlights multiple issues that deserve additional research. First, approximately 60.0% of patients interviewed reported PASC impacted their driving. This has important implications in Iowa and other rural states where healthcare barriers may include distance to specialist care and few public transportation options. Second, given their age range, many of our patients had children at home, with some also acting as caregivers to aging parents. Thus, the loss of their previous caregiver or provider role was two-fold and the impact on their household and extended family greater. Third, in their study on physical activity among COVID-19 survivors, Humphreys et al. speak of patients “adapting to an altered life,” and reckoning with the potential for permanent changes, an idea our patients emphasized as well [25]. Future research is needed in this area to assess how best to support COVID-19 survivors.

Other critical future avenues of research include addressing existing knowledge gaps in PASC by studying the underlying pathophysiology, for example, possible altered immunological function following viral infection, as well as the role of psychosocial factors leading to acute and long-term complications after SARS-CoV-2 infection. It is important to develop standardized diagnostic criteria to identify underlying risk factors, so that the condition can be prevented and treated in the future. Finally, it would be helpful for future research to examine potential differences in PASC experiences by other factors like gender, race or ethnicity, and employment status.

This study provides insights into patients’ struggles with PASC and highlights how they perceive the effects of brain fog and fatigue on their lives. Sharing these experiences may help clinicians and policy makers to better understand the patient perspective and offer adequate support to this patient population. Our study has certain limitations. This was an exploratory study with a small sample size. Study participants were also predominantly female, white, 40–55 years of age, and symptomatic at the time of participation. The purposive sampling strategy and sample size were appropriate choices, given available resources and recommendations by experienced qualitative researchers, and we have described the strengths of this approach [54,55]. However, our results may not be generalizable to a broader population. Participants also were insured (i.e., private insurance or Medicare) and had access to a tertiary healthcare facility with a dedicated PASC clinic, and thus possibly had increased symptom severity and/or were in greater distress compared to patients who did not seek care for their symptoms. 

## 5. Conclusions

Our findings highlight the negative impact of PASC symptoms, including brain fog and fatigue, on daily activities, driving, work/employment, and interpersonal relationships. Important related themes included the relapsing-remitting nature of symptoms, synergistic effects of brain fog and fatigue, difficulty with multitasking, feelings of isolation, a sense of loss, negative self-perception, and underlying social stigma. This study provides insight into patients’ descriptions of their recovery process, as well as the perceived effects of neurocognitive symptoms and fatigue. These insights may help researchers to develop future studies to address the knowledge gap in PASC and help clinicians to understand patient perspectives to ensure the delivery of patient-centered healthcare. It is important to be sensitive and timely in identifying the needs of COVID-19 survivors to better support them during their recovery. 

## Figures and Tables

**Table 1 ijerph-19-15499-t001:** Characteristics of interview participants.

Characteristics	Participants (*n* = 15)
Age (years)	
Average (range)	49.3 (40–68)
Sex, *n* (%)	
Female	10 (66.7)
Male	5 (33.3)
Race, *n* (%)	
White	13 (86.7)
Hispanic/Latino (of any race)	2 (13.3)
Marital Status, *n* (%) *	
Married	11 (73.3)
Divorced	4 (26.7)
Employment status, *n* (%) **	
Employed	8 (53.3)
Unemployed	6 (40.0)
Retired ***	1 (6.7)
Insurance Status, *n* (%) *	
Private insurance	14 (93.3)
Medicare, with secondary insurance	1 (6.7)
Duration of Symptoms (months) ****	
6–8	2 (13.3)
9–11	6 (40.0)
12–14	3 (20.0)
15–17	2 (13.3)
18–20	2 (13.3)
Hospitalization, *n* (%)	
Ambulatory *****	9 (60.0%)
Admitted to hospital	4 (26.7%)
Admitted to hospital and to ICU	2 (13.3%)

* At time of COVID-19 diagnosis, per chart review; ** At time of interview per interview data; *** Retired before the COVID-19 pandemic; **** Defined as persistent symptom(s) since time of COVID-19 diagnosis measured at time of interview; ***** Not admitted to hospital; ICU: intensive care unit.

## Data Availability

Not applicable.

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
