# Peer review of "Brain Fog and Fatigue following COVID-19 Infection: An Exploratory Study of Patient Experiences of Long COVID"

_ijerph, 2022, doi:10.3390/ijerph192315499_

Round 1
Reviewer 1 Report
Thank you for the opportunity to review the manuscript “Brain Fog and Fatigue following COVID-19 infection: Patient Experiences of Long COVID.”
Strenghts: The paper has reviewed existing evidence and compares with the current ones. Also, the methods section clearly describes the objective of the study and its analysis to understand the results.
Still there are some statements which require more attention from the authors:
- The introduction is poorly writen, needs rephrasing and new information introduced (with quotations);
- Small sample size;
- Please explain line 86-89, how is the data coded? What algoritm did you used?; please see: Saldana, Johnny (2009). The Coding Manual for Qualitative Researchers. Thousand Oaks, California: Sage.
- 364-365- there are other conditions that resonates with PASC..
- see also The Lancet Neurology. “Long COVID: understanding the neurological effects.” The Lancet. Neurology vol. 20,4 (2021): 247. doi:10.1016/S1474-4422(21)00059-4;
Reviewer 2 Report
To the authors:
This article focuses on better understanding the lived experiences of patients who have suffered long-term COVID-19 by concentrating on some characteristic syndromes such as fatigue or brain fog.
The conduct of qualitative studies is a handicap for the authors given that most scientific studies are quantitative, we can say that this is a contribution of the work.
The first thing that is missing is that the introduction is so short, from lines 38 to 62, the “state of the art” part in only 10 lines (38-48), we believe that a more exhaustive presentation should be made. The methodological part of the COVID-19 disease cannot be presented in only 14 lines (49-62).
In the methodological part, we must emphasize that, at a general level, the work shows a problem of representativeness, as authors have shown throughout the text. As it deals with such a small sample, which is therefore difficult to extrapolate to the general population, i.e., it would have a serious issue of ecological validity.
Given this difficulty, although the authors comment on line 341, the title should include “exploratory study”, pilot study, or straightforward approach, as a way of minimizing the concern of the sample. We should also highlight that one of the deficiencies of the work is the lack of concreteness in the use of the MAXQDA qualitative analysis software, which allows the creation of different figures such as “word cloud” or frequency analysis. Although the attached document shows some sentences emitted by the 15 patients, the text should clearly express the “codes used”.
The frequencies of these codes must be shown, beyond presenting an overview of the responses that do not allow us to have an individual view of them.
We believe that the answers given by the subjects should be better exploited, establishing categories and codes, and presenting a table of frequencies of these words. As in other qualitative analysis programs, such as ATLAS.TI, NVIVO, or IRAMUTEQ, it is very important to know how many times the subjects highlight a word in the lexicometric analysis.
It is necessary to describe how the transcription was carried out after the recording of the patients, by means of voice recognition software or manual transcription by the research team. How many questions were included in the semi-structured telephone interview: did the questions ask to elicit possible symptoms, or were they open questions, where patients expressed their opinion, ruling out social desirability bias?
It would be interesting to know whether the sample of working patients (8) versus unemployed and retired patients (7) had significant differences in the impact on personal relationships. Also, it could be analyzed whether there were significant differences in the 10 women between those who were working and those who were unemployed in 3.3 and 3.5. Finally, I think that the authors should make a little more effort to present more comprehensive final conclusions (seven lines). It would be useful to present guidelines for future research and extend them to studies where long COVID-19 can be contrasted with short COVID-19 and find common patterns of behavior in both cases.
Round 2
Reviewer 1 Report
I have no further comments. Thank you !
Author Response
We thank the Reviewer 1 for the comments.